# Insights into the Effects of Hydroxycinnamic Acid and Its Secondary Metabolites as Antioxidants for Oxidative Stress and Plant Growth under Environmental Stresses

**Sindiswa Khawula [1], Arun Gokul [2], Lee-Ann Niekerk [3], Gerhard Basson [3], Marshall Keyster [3], Mihlali Badiwe [4], Ashwil Klein [5] and Mbukeni Nkomo [1,\*]**

[1] Plant Biotechnology Laboratory, Department of Agriculture, University of Zululand, Main Road, Kwa-Dlangezwa 3886, South Africa; 201625768@stu.unizulu.ac.za

[2] Department of Plant Sciences, Qwaqwa Campus, University of Free State, Phuthadithaba 9866, South Africa; gokula@ufs.ac.za

[3] Environmental Biotechnology Laboratory, Department of Biotechnology, University of the Western Cape, Bellville 7535, South Africa; leeannniekerk@gmail.com (L.-A.N.); 3373827@myuwc.ac.za (G.B.); mkeyster@uwc.ac.za (M.K.)

[4] Department of Plant Pathology, Stellenbosch University, Stellenbosch 7435, South Africa; badiwem@sun.ac.za

[5] Plant Omics Laboratory, Department of Biotechnology, University of the Western Cape, Bellville 7535, South Africa; aklein@uwc.ac.za

\* Correspondence: nkomom@unizulu.ac.za; Tel.: +27-31-902-6170

**Abstract:** Plant immobility renders plants constantly susceptible to various abiotic and biotic stresses. Abiotic and biotic stresses are known to produce reactive oxygen species (ROS), which cause comparable cellular secondary reactions (osmotic or oxidative stress), leading to agricultural productivity constraints worldwide. To mitigate the challenges caused by these stresses, plants have evolved a variety of adaptive strategies. Phenolic acids form a key component of these strategies, as they are predominantly known to be secreted by plants in response to abiotic or biotic stresses. Phenolic acids can be divided into different subclasses based on their chemical structures, such as hydroxybenzoic acids and hydroxycinnamic acids. This review analyzes hydroxycinnamic acids and their derivatives as they increase under stressful conditions, so to withstand environmental stresses they regulate physiological processes through acting as signaling molecules that regulate gene expression and biochemical pathways. The mechanism of action used by hydroxycinnamic acid involves minimization of oxidative damage to maintain cellular homeostasis and protect vital cellular components from harm. The purpose of this review is to highlight the potential of hydroxycinnamic acid metabolites/derivatives as potential antioxidants. We review the uses of different secondary metabolites associated with hydroxycinnamic acid and their contributions to plant growth and development.

**Keywords:** hydroxycinnamic acid; hydroxycinnamic acid derivatives; environmental stresses

## 1. Introduction

Environmental stresses include non-biotic (nonliving, e.g., extreme temperatures, drought, flood, light, salt, viruses, and heavy metals) and biotic (living, e.g., insects and microorganisms such as bacteria) factors that have a great potential to cause serious modifications to the plant's natural environment, thus resulting in secondary stresses. Examples of secondary stresses includes osmotic stress and oxidative stress [1]. Oxidative stress occurs due to an imbalance in ROS molecules such as the superoxide radical ($O_2^-$), hydrogen peroxide ($H_2O_2$), and hydroxyl radical ($OH\cdot$), triggering a disturbance to plants' physiological and metabolic processes [2]. Under normal conditions, ROS molecules are

produced at low levels, and play an essential role in plant intracellular redox signaling regulating plant growth and development [3]. It is vital to control the levels of ROS molecules, as exposure of plants to biotic and non-biotic factors results in an increase in ROS, which may induce damage to proteins, lipids, and DNA molecules [4]. To regulate ROS, plants employ antioxidant defense mechanisms that detoxify ROS, reducing the oxidative damage to protein, lipid, and DNA molecules. This aids in the prevention of programmed cell death and subsequent yield loss [5]. The basic antioxidant system employed by plants includes enzymatic defense mechanisms such as ascorbate peroxidase, catalase, glutathione reductase, glutathione S-transferase, guaiacol peroxidase, and superoxide dismutase [4,6]. In addition to enzymatic defense mechanisms, plants also possess a non-enzymatic antioxidant system such as ascorbate or glutathione that also plays an important role in the protection against oxidative damage [7].

Furthermore, plants also possess phenolic compounds, which play crucial roles in regulating diverse physiological processes in plants such as growth and development, and in mediating plant responses to biotic and non-biotic stresses [8,9]. Phenolic compounds are secondary plant metabolites that are abundant in foods and are influenced by the plant age, genotype, tissues, season, and exposure time to stress [10]. These phenolics perform many biological functions such as being a structural component of cell walls and regulating auxin transport and the chalcone synthase gene [10–12]. The biosynthesis of phenolic compounds involves various enzymes such as phenylalanine ammonia-lyase (PAL) and phosphoenolpyruvate (PEP)-carboxylase [13]. The structural makeup of plant phenolics can be divided into two categories: cinnamic acid derivatives (hydroxycinnamic) and benzoic acid derivatives (hydroxybenzoic), both of which are signaling molecules that regulate the expression of antioxidant genes and play a role in the maintenance of cellular redox homeostasis in plants via ROS toxicity regulation. While much is still being learned about the signaling interaction between phenolic acids and antioxidant defense systems, significantly less is known about the networking and regulation of their biosynthesis under biotic and non-biotic stresses. In particular, the relationship between the enzymes responsible for the phenylpropanoid compounds and their potential inhibitor has not been extensively reported. In this review, we highlight current studies on the diversity of hydroxycinnamic acid metabolites/derivatives as well as the possible role of enzyme inhibitors involved in the formation of these metabolites. We further explore the potential technologies that might be used as the main approaches for understanding the mechanism of the interactions between these derivatives and ROS homeostasis. The main focus of the discussion will be on the influence of phenolic acids involved in the phenylpropanoid pathway and their role under stressful conditions.

## 2. Phenolic Acid Diversification

Plant phenolic compounds are an important class of secondary metabolites that are ubiquitous in nature and are found from plants to fungi [14]. They can be grouped into three main categories, namely phenolics, flavonoids, and tannins, commonly referred to as polyphenols [15]. However, the scope of this review will focus on phenolic acids, which are mostly referred to as monophenols because they contain only one phenol ring [16]. These phenolic acids are mostly found in the outer layers of most grains in both free and bound forms, where they are thought to inhibit the growth of microorganisms [17]. To date, more than 10,000 phenolic acids have been identified in various plant species [18], where they perform a broad array of protective functions, including antimicrobial, photoprotective, structure-stabilizing, and signaling functions in the promotion of plant growth [19]. They have also been exploited for several other applications including bioremediation and the production of allelochemicals and antioxidants [20]. Inside the plants, phenolic acids are believed to interact with external environmental stimuli in a complex signal transduction ranging from the roots to the leaves and to be involved in the mobilization of nutrients [21], while in humans, they have been linked to having a hyper-protective effect against diabetes [22] and lowering cholesterol [23]. Phenolic acids are readily

absorbed through the intestinal tract walls of mammals on a regular basis from vegetables, fruits, cereals, tea/coffee, and spices, which offer roughly 25 mg per day depending on the diet [24]. The main dietary sources of phenolic compounds are fruits and plant-derived beverages such as fruit juices, tea, coffee, and red wine [11]. Ripe fruits such as apples, various berries, plums, cherries, some citrus fruits, and peaches are the major dietary sources of phenolic acids. Foods such as cereals, carrots, salads, eggplants, cabbage, and artichoke are also rich in these compounds. Furthermore, food sources such as coffee beans and olives are among the richest sources of caffeic acid [25]. Tea is also a common source of hydroxycinnamic acids such as caffeic, *p*-coumaric, and chlorogenic acids. During tea fermentation, some phenolic compounds, including 5-CQA, are oxidized, changing the tea's color, taste, scent, and aroma in a way that profoundly affects the quality of the drink [25]. Plant foods (including fruits, cereal grains, legumes, and vegetables) and beverages (including tea, coffee, fruit juices, and cocoa) are major sources of phenolics in the human diet [19].

### 3. Phenolic Acid Generation in Plants

Phenolic acids can be synthesized endogenously in plants via enzymatic pathways such as the pentose phosphate, shikimate, and phenylpropanoid pathways (PPP). The production of hydroxycinnamic acid metabolites begins with phosphoenolpyruvate and erythrose-4-phosphate. The shikimate route then generates phenylalanine, which triggers phenyl deamination by PAL [26], while phenylalanine is first converted into cinnamic acid by phenylalanine ammonia (PAL) that is further converted into the *p*-coumaric acid cinnamate 4-hydroxylase ($C_4H$) enzyme [27]. As part of the phenylpropanoid pathway, *p*-coumaric acid is also a precursor of other phenolic acids such as caffeic, ferulic, and sinapic acids (see complete schematic details in Figure 1A). The enzyme cinnamate-4-hydroxylase ($C_4H$) hydroxylates trans-cinnamic acid by adding a hydroxyl group to position 4 to create *p*-coumaric acid. The enzyme coumaryl 3-hydroxylase ($C_3H$), a necessary follow-up step in forming hydroxycinnamic acids, adds a hydroxyl group to position 3 to create caffeic. The enzyme caffeic acid O-methyltransferase (COMT) converts caffeic acid into ferulic acid by 3-O-methylating it. The enzyme 4-coumarate CoA ligase joins one coenzyme A (CoA) with caffeic acid, causing the synthesis of chlorogenic acid, which is made when caffeoyl-CoA is esterified with quinic acid by the enzyme hydroxycinnamoyl-coenzyme A quinate transferase (HCQT). This is the final stage in the biosynthesis of hydroxycinnamic acid, and sinapic acid is produced from ferulic acid by its hydroxylation at position 5 and subsequent O-methylation through the actions of ferulic 5-hydroxylase (F5H) and COMT, respectively. *p*-Coumaric, ferulic, and sinapic acids can be directed to lignin production. Specifically, the phenolic acids produced by the phenylpropanoid pathway are indispensable to plants because they are mostly required as a starting point to produce other compounds, such as flavonoids, coumarins, and lignans [28].

Although much is understood about the diversity and accumulation of the phenylpropanoid products, less is understood about the networking and control of their biosynthesis. In particular, the relationship between the enzymes responsible for the phenylpropanoid compounds and their potential inhibitor has not been extensively reported. Although research on the potential inhibitors of these phenolic acids remains limited, Schalk et al. showed that of the 16 potential inhibitors of cinnamate 4-hydroxylase, only $C_4H$ was tested. While 3,4-(methylenedioxy)-phenyl acetic acid (15%) and methylenedioxy-aniline (13%) together with ayapin (13%) and piperonyl chloride (13%) showed a slight inhibition towards the $C_4H$ enzyme, piperonylic acid (PA) showed the most prominent results with 58% inhibition. A more in-depth approach was also proposed by the authors in [29], who determined that knocking down the $C_4H$ enzyme resulted in the overproduction of trans-cinnamic acid, which leads to the biosynthesis of benzoic acid (also indicated in Figure 1B,C). Benzoic acid also plays a major role in plant growth and development, through the regulation of photosynthetic activity and transpiration rate together with ion uptake and transportation [30]. In contrast to hydroxycinnamic acids,

hydroxybenzoic acids can also be produced directly from the shikimic acid pathway. Since they are not phenylpropanoids, they can still be generated even in the absence of PAL. Among the most significant hydroxybenzoic acids are gallic and salicylic acids. According to studies, the shikimate dehydrogenase enzyme converts 3-DHS into 3,5-didehydroshikimate, which is then used to create gallic acid in the shikimic pathway. This latter substance tautomerizes to produce gallic acid, its redox counterpart. The formation of polymeric compounds like gallo- and ellagitannins is facilitated by gallic acid. The shikimic acid pathway allows for the synthesis of salicylic acid in plastics.

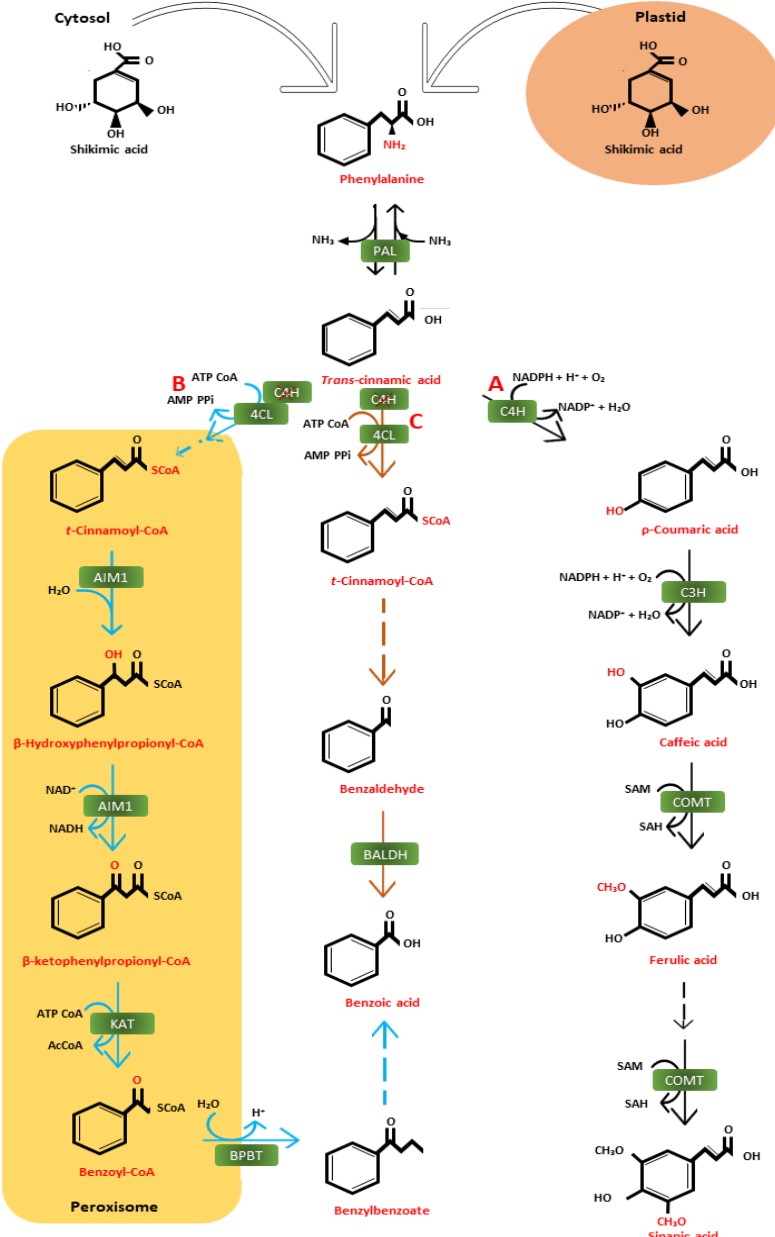

**Figure 1.** A schematic diagram representing the biosynthesis pathway from phenylalanine to either hydroxycinnamic acids or hydroxybenzoic acids. Phenylalanines derived from shikimic acids, coming from either the cytosol or plastid, are utilized to biosynthesize hydroxycinnamic acids (in the cytosol) (**A**). However, a divergence occurs in this biosynthetic pathway when there is limited to no $C_4H$ to convert trans-cinnamic acid into ϱ-coumaric acid. This divergence leads to the CoA-activation of trans-cinnamic acid by 4CL, to flow into either the (**B**) β-oxidative (peroxisomal) or (**C**) non-oxidative production hydroxybenzoic acids. Solid arrows signify established biochemical reactions, while dashed arrows signify hypothesized steps or steps not yet identified. Black arrows follow the

hydroxycinnamic acid biosynthetic pathway, orange arrows follow the the non-oxidative hydroxybenzoic acid biosynthetic pathway, and blue arrows follow the β-oxidative hydroxybenzoic acid biosynthetic pathway. Enzymes are indicated in green blocks and are abbreviated as follows: PAL (phenylalanine ammonia lyase), C4H (cinnamate 4-hydroxylase), C3H (ǫ-coumarate 3-hydroxylase), COMT (caffeoyl O-methyl transferase), 4CL (4-coumarate:CoA ligase), AIM1 (abnormal inflorescence meristem 1), KAT (3-ketoacyl-CoA thiolase), BPBT (benzoyl-CoA:benzylalcohol/2-phenylethanol benzylalcohol), and BALDH (benzaldehyhde dehydrogenase) [29,31–33].

## 4. Phenolic Acids in Plants under Salt Stress Conditions

The role of phenolic acids in controlling plant responses to abiotic stress has received considerable attention, with findings demonstrating that phenolic acids and their derivatives are rapidly triggered by a variety of stressors such as salinity (NaCI) stress [11,34,35]. The accumulation of NaCI concentration can increase oxidative stress by directly producing ROS molecules such as $H_2O_2$, which can trigger plant programmed cell death [36]. As a result, maintaining low levels of $H_2O_2$ is important for cell tolerance to environmental stress. Figure 2 hypothetically illustrates the impact of salt (NaCI) upon entering the cell, where red arrows indicate NaCI-induced oxidative damage and green arrows illustrate the HCA-induced alleviation of NaCI-induced $H_2O_2$ accumulation damage. $H_2O_2$ induces signaling responses as well as signal molecules such as calcium ($Ca^{2+}$), salicylic acid (SA), abscisic acid (ABA), jasmonic acid (JA), ethylene, and nitric oxide (NO), functioning together in signal transduction pathways to mediate responses to environmental resistance and regulate plant growth and development. These concentrations of wall-bound phenols may serve as a source of phenylopropanoid units for lignin biosynthesis or perhaps be the start of the lignification process itself [8]. Because of the abundance of bonded molecules in these esters, light energy is transferred, causing changes to the plant cell wall structure, water flow, tugor pressure, and growth [8,21].

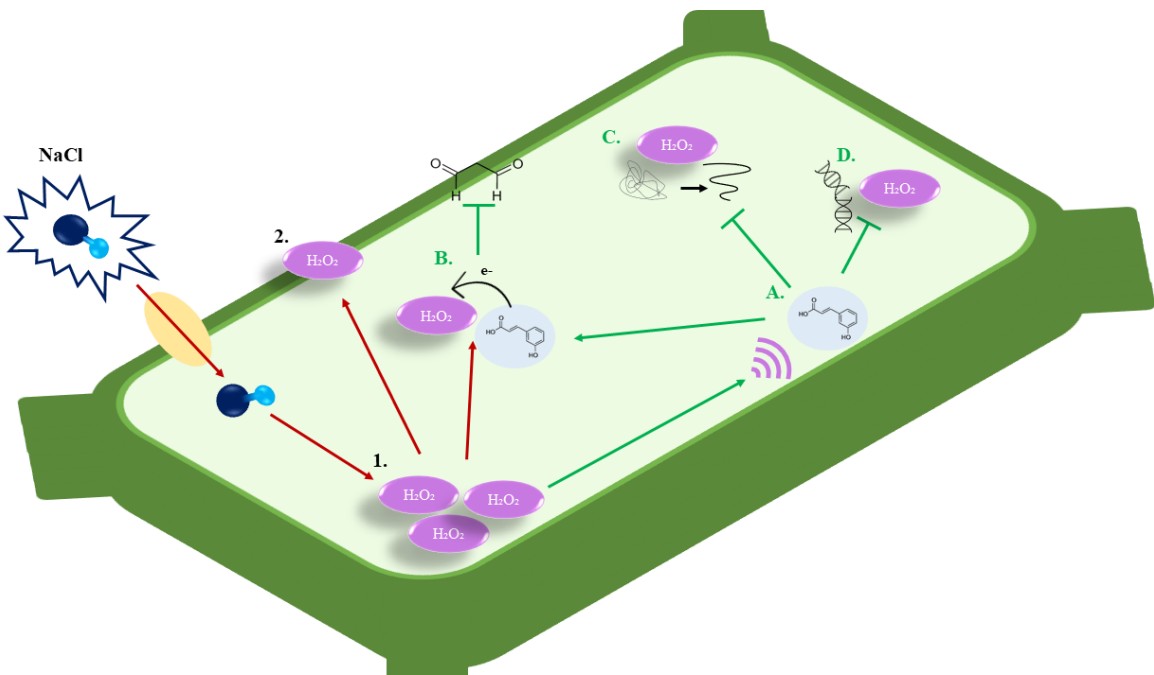

**Figure 2.** A schematic diagram presenting the mechanism of hydroxycinnamic acid (HCA) towards salt-induced oxidative damage. Under salt stress, sodium chloride (NaCl) enters the plant cell, where once inside, it (**1**) induces hydrogen peroxide ($H_2O_2$) accumulation. If not controlled, NaCl-induced $H_2O_2$ accumulation results in (**2**) peroxidation of the plant cell membrane, leading to extreme damage and even cell death. However, multiple studies have documented that HCAA is a response to salt stress. (**A–D**) The mechanism of HCA in reducing salt-induced oxidative damage in plant cells. (**A**) The salt-induced accumulation of $H_2O_2$ has been shown to increase the production of HCAs. (**B**) These HCAs then reduce cell membrane peroxidation, by donating an electron to $H_2O_2$,

reducing the $H_2O_2$-induced lipid peroxidation of the cell membranes. (**C,D**) HCAs have likewise been documented to possess other antioxidant activities that reduce the occurrence of other types of oxidative stress damage, such reduced salt stress-induced protein degradation and DNA damage. Red arrows illustrate the salt-induced oxidative damage pathway and green arrows illustrate the HCA-induced alleviation of salt-induced $H_2O_2$ accumulation damage.

A study by [37] observed an increase in salt tolerance in *P. giganteum* treated with salicyclic acids (hydroxybenzoic acid). The authors also attributed the increase in salinity tolerance to an increase in chlorogenic acid (hydroxycinnamic acid derivative) in the plants treated with salicylic acid under salinity stress. The interplay between the different phenolic groups such as the hydroxybenzoic acids and hydrocinnamic acids presents an important avenue to explore in trying to understand their role in salt tolerance in plants. Additionally, a study by Gupta and De [38] observed the differences in phenolic compound production in rice cultivars with contrasting salinity tolerance. In their study, the authors observed a shift in metabolic processes in tolerant rice cultivars, shifting from producing cinnamic acid, a hydroxycinnamic acid, to gentisic acid, a hydrobenzoic acid, thus leading to the conclusion that the ability to induce these metabolic shifts could be used as a biomarker for salt tolerance in rice. This was further supported by the observation that in salt-sensitive rice cultivars, the metabolic pathways were more geared towards producing hydroxycinnamic acids. These findings, however, oppose some observations made in other studies whereby an increase in hydroxycinnamic acid content improved salt tolerance in plants. Linić et al. [39] tested three Brassica crops for their interaction with salt; the observation included kale and white cabbage having a higher tolerance while also having a higher proportion of hydroxycinnamic acids to hydrobenzoic acids. In a study by Nedved et al. [40], the authors observed the role of chitosan and hydroxycinnamic acid conjugates (chitosan: ferulic acid and chitosan: caffeic acid) on the growth and antioxidant activity in seven-day-old cucumber (*Cucumis sativus* L.) seedlings, under salt stress. The study highlighted the ability of exogenous chitosan–HACC conjugates to either stimulate growth, specifically the root and shoot length ratio, or curb oxidative stress, depending on the growth conditions [40]. Notably, under stress-free conditions, seed treatment with chitosan–ferulic acid (Ch-FA) and chitosan–caffeic acid (Ch-CA) indicated growth-stimulating effects on seedlings [40]. These seedlings exhibited no significant alterations in their antioxidant status relative to the control plants, thus leading to the assumption that the conjugates are effective as growth regulators in the absence of stress [40]. On the contrary, under salt stress conditions, no alterations were observed in their growth parameters relative to the control plants, except for that the increased root–shoot length remained. However, the seedlings pretreated with conjugates exhibited decreased lipid peroxidation due to increased peroxidase activity under salt stress [40]. Thus, it was assumed that the conjugates can increase seedlings' resistance to prolonged salt stress because of a decrease in the intensity of oxidative processes [40]. The aforementioned studies show the importance of understanding the phenolic profile, especially with a focus on hydroxycinnamic acids within different plant genera, as it has been shown to differentially affect their responses to abiotic stresses such as salinity.

## 5. The Impact of Hydroxycinnamic Acids and Their Derivatives on Drought Stress

One of the most significant issues limiting agricultural productivity is drought, which negatively impacts crop yield. Plants activate their drought-response mechanisms in response to water limitation, some of which include morphological and structural changes, the expression of drought-resistant genes, synthesis of hormones, and the production of chemicals involved in osmotic regulation [41]. It has been suggested that the accumulation of phenolic compounds is a well-known adaptive mechanism in the olive tree against water deficit [42]. To combat the subsequent over-accumulation of ROS because of water stress or possibly because of the decrease in lignin biosynthesis, a rise in phenolic compounds including HCAAs has been found in samples from water-stressed

plants. In a study by Torras-Claveriam et al. [43], the authors reported an increase in phenolic compounds, which included seven hydroxycinnamoylquinic acids, seven hydroxycinnamic acid glucosides, and three hydroxycinnamic acid amides in water-stressed samples in relation to control plants. Furthermore, Latif et al. [44] reported a decrease in the negative physiological effects of drought stress on *Zea mays* growth parameters (shoot fresh weight, shoot dry weight, root fresh weight, root dry weight, root length, and root area) upon the exogenous application of hydroxcinnamic acid derivatives, namely salicyclic acid. In this study, the authors concluded that the buildup of both soluble and cell wall-bound phenolics caused by SA's foliar spray may influence maize's ability to withstand drought stress. The abovementioned results highlight the importance of comprehending how different phenolic compounds, and in particular hydroxycinnamic acid, affect plant responses to abiotic stresses such as drought at the species level.

## 6. The Role of Hydroxycinnamic Acids and Their Derivatives in Heavy Metal Tolerance

Heavy metals have been observed entering estuarine and agricultural systems through industrial sources [45]. Many plants display stress responses such as chlorosis as the heavy metal concentration increases within the environment. However, certain plants were observed to possess an innate tolerance to these heavy metals [46]. The proposed reason for this tolerance was associated with phenolic acids [47]. In a study by Chen et al. [48], the authors observed the role of phenolic acid content in ROS homeostasis and the bioavailability of Cd and Zn. The study showcased the ability of the phenolic group of hydroxycinnamic acids and their derivatives to activate heavy-metal defense mechanisms in *K. obovata*. Furthermore, the mechanism of heavy-metal defense by plant secondary metabolites includes the regulation of ROS and a free radical scavenging ability in conjunction with the enzymatic antioxidants. The importance of the hydroxycinnamic acid group under Cd and Zn treatment was further highlighted as their concentrations (chlorogenic, cinnamic, and ferulic) were reported to be higher than the other phenolic compounds observed. The same phenomenon was observed in *Z. mays*, where chlorogenic acid and rutin were identified as the primary phenolics within the leaves, and the addition of Cd and Pb led to an enhancement in total phenolic content [49]. It should be noted that both the composition of hydroxycinnamic acids and their concentration are important, as this may dictate which defense mechanisms are employed. The second mechanism identified by Chen et al. [48] in *K. obovata* included the chelation of the metals through cinnamic and coumaric acids. The mode of action of the hydroxycinnamic acids includes the use of functional groups such as the carboxyl and hydroxyl groups that bind the heavy metals [47]. Chen et al. [48] also proposed an alternative mechanism, which was the secretion of chlorogenic, cinnamic, and coumaric acids from the root into the sediment, to create adverse conditions for heavy-metal uptake. A study by Lwalaba et al. [50] observed increases in *p*-coumaric and ferulic acids when *H. vulagare* cultivars were exposed to Co and Cu. Their study highlighted the potential of these compounds to use hydrogen peroxide to polymerize monolignols to form lignin, which forms natural barriers as well as enhances metal sequestration into cell constituents such as the cell wall. The findings in the aforementioned studies spotlight the numerous ways in which the hydroxycinnamic acid group is mobilized in order to improve plants' defenses against heavy metals.

## 7. The Role of Hydroxycinnamic Acids and Their Derivatives in Plant Disease Resistance

Evolutionary progression has enabled plants to develop numerous mechanisms to combat pathogen proliferation and restrict severe infection. Proficient recognition of pathogens initiates pathogen-associated molecular pattern (PAMP)-triggered immunity (PTI) by the induction of defense gene expression. Some of the defense enzymes associated with plant resistance to pathogen infection include β-1,3 glucanase, peroxidase (PO),

polyphenol oxidase (PPO), and phenylalanine ammonia-lyase (PAL) [51,52]. PAL is a commencement enzyme of the phenylpropanoid pathway, which is intrinsically tied to the synthesis of lignin, phytoalexins, and phenolic compounds. These compounds are secreted to serve as protection agents against herbivores, insects, fungi, bacteria, and viruses [53–55]. The largest class of phenolic acids is the hydroxycinnamic acids (HCAs), which consist of caffeic, ferulic, *p*-coumaric, and sinapic acids [56]. HCAs may occur in derivative forms, such as amides (in combination with amino acids or peptides) and esters (in combination with hydroxyl acids or glycosides), which play an important protective role in plant cells against pathogen attacks by strengthening plant cell walls and acting as antimicrobial agents [57–59]. Caffeoylserotonin, cinnamoyltyramine, coumaroylserotonin, coumaroylputrescine, feruloylagmatine, feruloylserotonin, feruloylputrescine, and cis-pcoumaroylagmatine (HCAAs) may be transported to the cell wall by glutathione-S-transferase (GST) during pathogen infection to mitigate the infectious spread [60]. Furthermore, ROS production is a key indicator of the successful identification of an infection, which initiates the activation of defense mechanisms in plants. HCAA-induced $H_2O_2$ is implicated in the initiation of plants' hypersensitive responses (HRs), which induce localized host cell death and prevent further pathogen infection. Therefore, $H_2O_2$ is utilized during cell-wall lignification for enhanced rigidity, to achieve cell protection against pathogen penetration [61]. Plants can also utilize their dynamic stomatal system as part of their innate immunity. Upon successful pathogen detection, the biosynthesis of endogenous abscisic acid (ABA) increases and interacts with HCAAs to regulate the stomatal opening and closing in response to the pathogen invasion [62]. The membrane-bound protein MATE (multidrug and toxic compound extrusion) also plays a pivotal role in the defense systems of plants. HCAAs are transported from the cytoplasm into various plant tissues to accumulate antimicrobial activity at the infected sites [62].

A high content of phenolic compounds has been associated with increased resistance to pathogen infection in plants. Ghorbi et al. [63] evaluated *Pyrenophora tritici-repentis* (Ptr) infection (causing Tan spot) in wheat and demonstrated a substantial accumulation of chlorogenic acid, ferulic acid, *p*-coumaric acids, rutin, and vanillic acid in resistant and semi-resistant cultivars. Examination of the interrelations between two viticultural diseases (powdery and downy mildew diseases) and phenolic changes in tolerant cultivars revealed increased catechin, epicatechin, and gallic acid contents after infection with powdery mildew disease [64]. Feruloylputrescine (phenolic amide derivative of ferulic acid and putrescine) was reported to be uniquely synthesized in tolerant wheat near-isogenic lines (NILs) after *F. graminerum* infection, illustrating that it is induced in a resistance-related manner [65]. Furthermore, upon the metabolomic examination of *Puccinia striiformis* f. sp. *tritici* infected wheat, *p*-coumaroyl agmatine in the hydroxycinnamic acid amide (HCAA) pathway was significantly upregulated after infection [62]. This is a phenolic amide that is formed through the conjugation of *p*-coumaric acid with agmatine, and it possesses antioxidant and antimicrobial activity. The HCAAs pathway is a prominent subsidiary pathway of the phenylpropanoid pathway, and it contributes towards plant cell protectants against pathogenic attacks by strengthening cell walls. HCAAs are synthesized in the cytosol and then channeled into the cell wall through a peroxidase-mediated process in response to pathogen invasion [66,67] (See Figure 3).

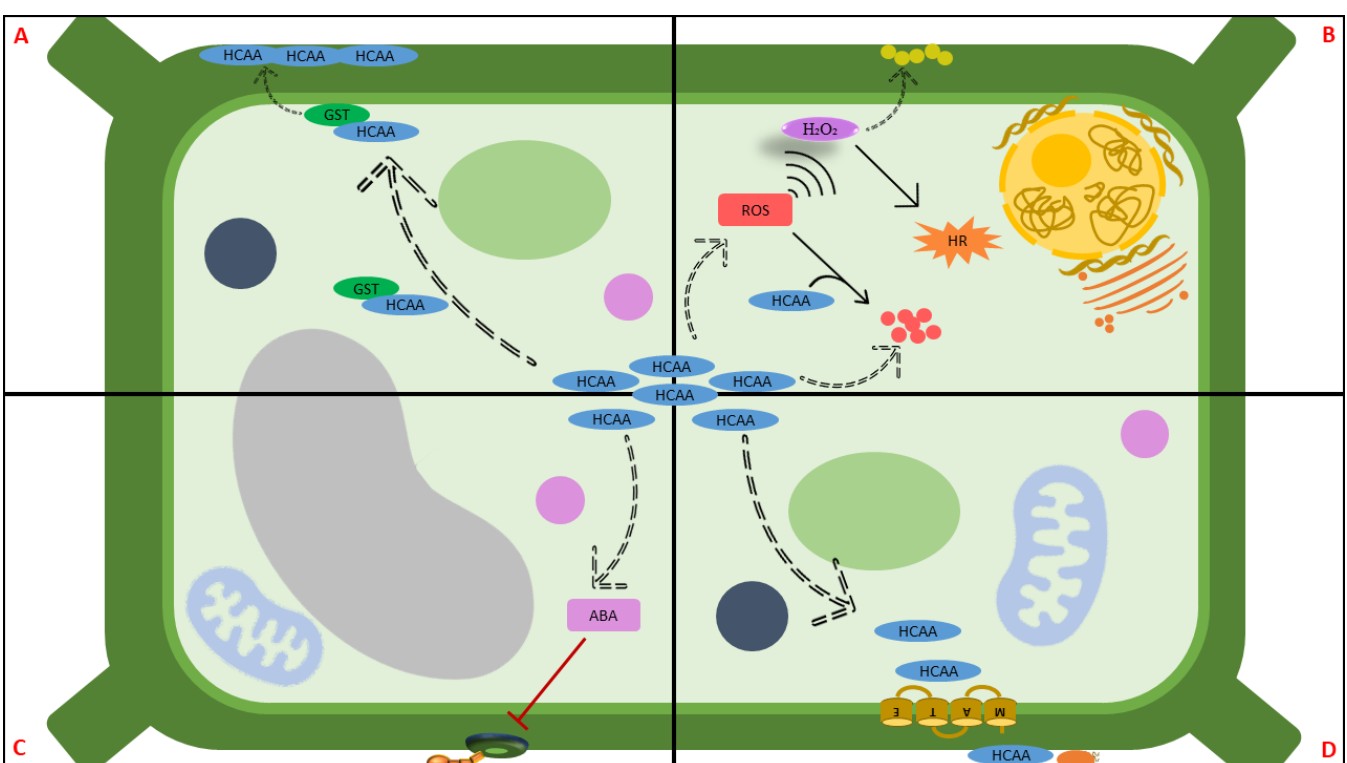

**Figure 3.** A schematic diagram depicting the mechanisms behind hydroxycinnamic acid amide (HCAA) pathogenic suppression. HCAA accumulation has been revealed as a common response to pathogenic infection; however, there are multiple avenues through which HCAA protects the plant cells. (**A**) A cell-wall strengthening approach, in which HCAAs with the assistance of glutathione S-transferase (GST) are translocated to the plasma membrane, and once at the plasma membrane, the HCAAs can be deposited (and accumulated) on the cell wall. (**B**) A ROS-mediated approach, in which an accumulation of HCAAs controls cellular ROS levels. HCAAs thus preserve redox homeostasis by being a resource for ROS biosynthesis, while simultaneously serving as radical-scavenging compounds, and in this way preserve redox homeostasis. (**C**) An ABA-mediated response, in which HCAA is implicated in inducing the ABA-regulation of stomatal opening and closing, to restrict pathogen invasion. (**D**) A direct attack, by which the multidrug efflux (MATE) transporter facilitates HCAA deposition. The image was generated from [61,62,68].

## 8. Role of Phenolic Acids in Plant Growth under Temperature Stress

When plants are exposed to cold temperatures, they undergo significant physiological changes, such as decreased photosynthesis and growth. This is due to temperature limits induced by the freezing point of plants' intracellular water and protein denaturation at higher temperatures, which limit plant life to a temperature range of roughly −10 to +60 °C [69]. This is with the exception for woody trees, which are well-adapted to intracellular water freezing, and plants in deserts, where daily temperatures reach 60 °C. According to Moreira et al. [70], an increase in temperature resulted in a considerable rise in the concentration of phenolic compounds. The study also found that the type and quantity of phenolic compounds produced varied among different plant species, suggesting that the response to temperature changes is species-specific. Overall, the study by [70] suggested that rising temperatures associated with climate change may have significant implications for the production and quality of plants that are rich in phenolic compounds, with potential impacts on human health and nutrition. Furthermore, studies focused on the influence of hydroxycinnamic acids and their derivatives on cold stress, which has been well-reported. Jan et al. [2] studied the effects of cold stress on the accumulation of phenolic compounds in plants, concentrating on chlorogenic and ferulic acids in particular. This study did not directly apply exogenous phenolic acids to plants but rather measured the endogenous accumulation of these compounds in response to cold stress. This is in comparison

to Zhang et al.'s [71] study, which investigated the effect of exogenous ferulic acid on cold stress in cucumber seedlings. However, the two studies suggest that the accumulation of phenolic compounds, particularly hydroxycinnamic acids such as ferulic and chlorogenic acids, may be important for improving plant tolerance to cold stress and enhancing the total phenolic content in crops. The mechanisms of action used by phenolic acids such as ferulic and chlorogenic acids are not completely understood, but research suggests that these compounds are involved in enhanced stress tolerance through multiple pathways, including antioxidant activity, gene regulation, hormone signaling, and the modulation of phenolic biosynthesis pathways.

### 9. Integrating Proteomic Approaches to Improving Stress Tolerance

The "multi-omics" technologies include genomics, transcriptomics, proteomics, phenomics, and ionomics in delineating the complex molecular machinery and providing independent information about the genes, genomes, transcriptomes, proteomes, and metabolomes (phenols and ions/elements). Integrating these omic technologies has been proven to be successful in exploring the molecular mechanisms involving growth and development in response to environmental stresses in plants [72]. While genomic analysis has enhanced our understanding of transcriptomic changes in plants in response to environmental stresses, these transcriptomic changes are not always reflected at the protein level due to post-translation modification [21]. Hence, proteomics was established to deal with the large-scale expression of protein changes in any organ, tissue, or cell under various stress factors [73,74]. This proteomic approach offers the possibility to identify proteins associated with a particular environmental and/or development signal associated with a multitude of metabolites such as those of the phenylpropanoid pathway. Neilson and colleagues were among the first authors to shed more light on the identification of stress-related gene/proteins involved in regulating the biosynthesis and signaling of phenolic acids associated with the cell wall. This mechanism influences plant tolerance to environmental stresses [75]. Some of the phenolic-associated pathways identified in Neilson's study were later shown to be directly involved in the regulation of antioxidant enzymes, which are responsible for scavenging free radicals [76,77]. Kaur et al. [78] further demonstrated that the exogenous supply of HCA and its derivatives (*p*-coumaric, caffeic, ferulic, and sinapic acids) increased salt tolerance in two contrasting wheat cultivar seedlings [78]. These results were also corroborated by Jones and authors, when they reported a positive correlation between the exogenous application of phenolic acids and plant growth promotion under both favorable and unfavorable conditions [77]. Contrastingly, this contradicted the majority of previous research that suggested that phenolic acid inhibited the rate of seed germination and altered the root length and biomass of different species [79]. However, to ascertain if the observed results of an increase in plant growth under unfavorable conditions is as a result of phenolic acids (as indicated by [78] and Jones and co-workers [77]), a more recent study by Nkomo [80] used a proteomic approach to study the use of inhibitors such as piperonylic acid (PA; inhibits the $C_4H$ enzyme) and MDCA (inhibits the 4CL enzyme). The underlining theory behind Nkomo's study was that if the biosynthesis of these hydroxycinnamic acids is responsible for conferring stress tolerance, then due to the inhibition of the biosynthesis of these hydroxycinnamic acids under stressful conditions, it would be expected that plants would suffer even greater stress. However, observing Figure 1, it can still be argued that inhibiting the biosynthesis of hydroxycinnamic acid can also lead to stress tolerance, as shown in Figure 1, which shows that the inhibition of $C_4H$ leads to the production of salicylic acid, which numerous studies have shown to be involved in conferring environmental stress tolerance (e.g., Khan et al. [29]).

## 10. Future Perspectives

Despite significant research being undertaken on the accumulation and exogenous influence of HCAAs/HCCA-derivatives on plant tolerance to abiotic and biotic stress, the research results vary substantially, as the diversity and concentration of these compounds under (1) various stresses (i.e., abiotic vs. biotic), (2) in different plant species, and (3) in different plant tissues (i.e., shoots, stems, and leaves) remain a limiting factor to fully elucidating the mechanisms of the tolerance conferred by hydroxycinnamic acids. To address these variations, we propose the use of proteomic tools to investigate the mechanisms associated with HCAA-induced stress tolerance versus sensitivity by investigating changes in the functional gene products (i.e., proteins). This approach may provide more insight into the influence of HCCAs on the final molecular functional products, which could play a larger role in conferring stress tolerance. Furthermore, given the redundancy of the genetic code, comparative proteomics between different plant species treated with the same stress can be used in order to identify commonly differentially regulated proteins associated with a stress response (i.e., tolerance). In a similar manner, investigating the commonly regulated proteins between different species treated with the same HCCAs/HCCA-derivative would allow for the identification of proteins/pathways uniquely associated with HCCA responses in a manner independent from genetic differences, in order to develop a general understanding of the mechanism through which HCCAs protect against oxidative damage across a variety of different plant species.

## 11. Conclusions

In recent years, hydroxycinnamic acids (HCAAs) and their derivates have gained significant attention in the pharmaceuticals field due to their remarkable biological activities including UV protection, antioxidative properties, anti-inflammatory effects, and antibacterial qualities. This review delves into the generation and diversity of phenolic compounds present in plants, with a keen focus on HCAAs. The evidence discussed in the aforementioned studies highlights the vital roles of HCAAs in plant responses to both abiotic and biotic stresses. Our findings shed light on some of the various mechanisms through which HCAAs and their derivatives (HCCA derivatives) confer tolerance in plants.

These mechanisms include the regulation of processes involved in maintaining cell wall plasticity via the regulation ROS (i.e., $H_2O_2$) for lignification, or acting as precursors for monolignols during lignin biosynthesis. The ability to regulate cell wall plasticity has been proven to be a crucial trait in plant responses to heavy metals and salinity, as the cell wall serves as the initial barrier, regulating the mobilization of salts (i.e., ions) and heavy metals into plants. Phenols and their derivatives, namely cinnamic and coumaric acids, and the secretion of chlorogenic, cinnamic, and coumaric acids in plant roots can promote the chelation of metals in soil sediment. This presents another mechanism by which hydroxycinnamic acid derivatives improve plant heavy-metal responses through regulating ion uptake and reducing the degree of ion toxicity, a common effect of heavy-metal and salinity stress.

Furthermore, plants also accumulate phenolics under water stress, which influences the amount of soluble and cell wall-bound phenolics. These findings highlight the influence of hydroxycinnamic acids on cell-wall plasticity and their importance in regulating cellular augmentation to respond to changes in the osmotic gradient, which occurs under salinity and water stress.

In terms of disease tolerance, hydroxycinnamic acids and their derivatives have been shown to actively move from the cytoplasm to the plasma membrane to facilitate cell-wall strengthening. Additionally, these compounds exhibit antimicrobial activity. Hence, the translocation and incorporation of these compounds in cell walls assist with mitigating the spread of plant pathogens. Moreover, HCAAs influence ROS (reactive oxygen species) homeostasis, regulating ROS levels as free radicle scavengers, thus preserving the redox balance and preventing oxidative damage. Under temperature stress, the accumulation of

hydroxycinnamic acids and their derivatives has been reported to influence the regulation of genes, hormones, and antioxidants via ROS regulation to provide tolerance.

Hence, this study provides insight into the activities of HCCAs under various stresses and their role in protecting plants against oxidative damage, in addition to potentially increasing the quality of the plant products by increasing the concentrations of antioxidant compounds in edible plants.

**Funding:** This work was supported by the National Research Foundation of South Africa (NRF). This research received financial support from the NRF to M.N (Grant number: BAAP2204082813) A.G. (Grant number: 129493), and A.K. and M.K. (Grant numbers: 107023, 115280, 116346 and 109083). A.G. was also supported by the CRF fund of the University of the Free State.

**Data Availability:** There are no new data associated with this article.

**Conflicts of Interest:** The authors declare that there are no conflict of interest.

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
