# Peer review of "Insights into the Effects of Hydroxycinnamic Acid and Its Secondary Metabolites as Antioxidants for Oxidative Stress and Plant Growth under Environmental Stresses"

_cimb, doi:10.3390/cimb46010007_

Round 1
Reviewer 1 Report
Comments and Suggestions for Authors
The presented article concerns a very important topic. However, the article itself requires numerous corrections (some are indicated below). Another problem is the fact that in recent years comprehensive articles on hydroxycinnamic acid have been written:
"Coman, V., & Vodnar, D. C. (2020). Hydroxycinnamic acids and human health: Recent advances. Journal of the Science of Food and Agriculture, 100(2), 483-499.", "Liu, S., Jiang , J., Ma, Z., Xiao, M., Yang, L., Tian, B., ... & Yang, Y. (2022). The role of hydroxycinnamic acid amide pathway in plant immunity. Frontiers in Plant Science, 13, 922119."
The article requires corrections.
Title:
"as antioxidants against oxidative stress" - does this mean that they are antioxidants only under conditions of oxidative stress? Do they not have such properties under physiological conditions?
Abstract:
"Plant immobility renders them constantly susceptible to various abiotic and biotic stresses." - animal organisms are mobile and are also exposed to these factors.
"Phenolic acid can be divided into different subclasses" - maybe acids?
Introduction:
Line 37: Are viruses biotic or abiotic?
"Examples of secondary stresses includes osmotic stress and oxidative stress." - (include?) please add a reference to the literature
"antioxidant system such as ascorbate or glutathione" - maybe antioxidant compounds? Ascorbate is not a system.
"In this review, we will highlight current studies on the diversity of hydroxycinnamic acid metabolites" - To make it more "current", I recommend enriching the bibliography with newer items. 30/71 are from the last 5 years and 43/71 are from the last 10 years.
"Phenolic acids are readily absorbed through the intestinal tract walls of
mammals, on a regular basis from vegetables, fruits, cereals, tea/coffee, and spices, which offer roughly 25 mg per day depending on the diet chart." - please provide a reference to the literature
"Phenolic acids generation in plants" - this chapter lacks the latest reports, facts that have been known for years are described. This chapter may be shortened to the necessary minimum in favor of more current research.
Salt stress - no link to current research: "Nedved, E. L., Kalatskaja, J. N., Ovchinnikov, I. A., Rybinskaya, E. I., Kraskouski, A. N., Nikalaichuk, V. V., ... & Laman, N. A. (2022). Growth parameters and antioxidant activity in cucumber seedlings with the application of chitosan and hydroxycinnamic acids conjugates under salt stress. Applied Biochemistry and Microbiology, 58(1), 69-76."
Drought stress - "Sarker, U., & Oba, S. (2018). Drought stress enhances nutritional and bioactive compounds, phenolic acids and antioxidant capacity of Amaranthus leafy vegetable. BMC Plant biology, 18(1), 1-15."
Biotic stress "Macoy, D. M. J., Uddin, S., Ahn, G., Peseth, S., Ryu, G. R., Cha, J. Y., ... & Kim, M. G. (2022). Effect of hydroxycinnamic acid amides, coumaroyl tyramine and coumaroyl tryptamine on biotic stress response in Arabidopsis. Journal of Plant Biology, 1-11."
Please analyze the rest of the text in terms of current available literature.
Materials and methods - no section, it is not known how the research for the literature review was planned. What queries were performed in the past and what filters were used?
Moderate editing of English language required
Author Response
Resubmission of manuscript “Insights into the effects of hydroxycinnamic acid and its secondary metabolites as antioxidants for oxidative stress and plant growth under environmental stresses”
Thank you for the opportunity to revise our manuscript entitled ‘Insights into the effects of hydroxycinnamic acid and its secondary metabolites as antioxidants for oxidative stress and plant growth under environmental stresses’ We appreciate the careful review and constructive suggestions made by the two reviewers. It is our belief that the manuscript is substantially improved after making the suggested edits.
Following this letter we have addressed the reviewer comments with our responses in bold italics, including how and where the text was modified. Changes made in the manuscript are marked using track changes. The revision has been developed in consultation with all co-authors, and each author has given approval to the final form of this revision.
Thank you for your consideration and we look forward to your response.
Kind regards,
Dr. Mbukeni Nkomo
REVIEWER #1 COMMENTS:
Recommendation: Minor changes needed
Comments and Suggestions for Authors
The presented article concerns a very important topic. However, the article itself requires numerous corrections (some are indicated below). Another problem is the fact that in recent years comprehensive articles on hydroxycinnamic acid have been written:
"Coman, V., & Vodnar, D. C. (2020). Hydroxycinnamic acids and human health: Recent advances. Journal of the Science of Food and Agriculture, 100(2), 483-499.", "Liu, S., Jiang , J., Ma, Z., Xiao, M., Yang, L., Tian, B., ... & Yang, Y. (2022). The role of hydroxycinnamic acid amide pathway in plant immunity. Frontiers in Plant Science, 13, 922119."
We thank the reviewer for the comment, We have took the reviewers advice and added more recent related references that within or reviews. Below is the list of some of the papers that were incorporated into our review:
- Zaid, A., Ahmad, B. and Wani, S.H., 2021. Medicinal and aromatic plants under abiotic stress: a crosstalk on phytohormones’ perspective. Plant Growth Regulators: Signalling under Stress Conditions, pp.115-132.
- Prabhu, S., Molath, A., Choksi, H., Kumar, S. and Mehra, R., 2021. Classifications of polyphenols and their potential application in human health and diseases. Int. J. Physiol. Nutr. Phys. Educ, 6(1), pp.293-301.
- Rahman, M.M., Rahaman, M.S., Islam, M.R., Rahman, F., Mithi, F.M., Alqahtani, T., Almikhlafi, M.A., Alghamdi, S.Q., Alruwaili, A.S., Hossain, M.S. and Ahmed, M., 2021. Role of phenolic compounds in human disease: current knowledge and future prospects. Molecules, 27(1), p.233.
- Barros, J. and Dixon, R.A., 2020. Plant phenylalanine/tyrosine ammonia-lyases. Trends in plant science, 25(1), pp.66-79.
- Sharma, A., Sidhu, G.P.S., Araniti, F., Bali, A.S., Shahzad, B., Tripathi, D.K., Brestic, M., Skalicky, M. and Landi, M., 2020. The role of salicylic acid in plants exposed to heavy metals. Molecules, 25(3), p.540.
- Caldana, C., Degenkolbe, T., Cuadros‐Inostroza, A., Klie, S., Sulpice, R., Leisse, A., Steinhauser, D., Fernie, A.R., Willmitzer, L. and Hannah, M.A., 2011. High‐density kinetic analysis of the metabolomic and transcriptomic response of Arabidopsis to eight environmental conditions. The Plant Journal, 67(5), pp.869-884.
- Wu, X., Gong, F., Cao, D., Hu, X. and Wang, W., 2016. Advances in crop proteomics: PTMs of proteins under abiotic stress. Proteomics, 16(5), pp.847-865.
- Mustafa, G. and Komatsu, S., 2021. Plant proteomic research for improvement of food crops under stresses: a review. Molecular Omics, 17(6), pp.860-880.
- Uddin, M.N., Robinson, R.W., Buultjens, A., Al Harun, M.A.Y. and Shampa, S.H., 2017. Role of allelopathy of Phragmites australis in its invasion processes. Journal of Experimental Marine Biology and Ecology, 486, pp.237-244.
- Ahmad, R., Hussain, S., Anjum, M.A., Khalid, M.F., Saqib, M., Zakir, I., Hassan, A., Fahad, S. and Ahmad, S., 2019. Oxidative stress and antioxidant defense mechanisms in plants under salt stress. Plant abiotic stress tolerance: Agronomic, molecular and biotechnological approaches, pp.191-205.
- Kumar, N. and Goel, N., 2019. Phenolic acids: Natural versatile molecules with promising therapeutic applications. Biotechnology reports, 24, p.e00370.
- Arfaoui, L., 2021. Dietary plant polyphenols: Effects of food processing on their content and bioavailability. Molecules, 26(10), p.2959.
The article requires corrections.
Title:
"as antioxidants against oxidative stress" - does this mean that they are antioxidants only under conditions of oxidative stress? Do they not have such properties under physiological conditions?
We thank the reviewer for the comment. We further support his statement that antioxidants are not only limited to oxidative stress, they play a crucial role in maintaining overall health of the plants under both normal and stressful conditions (protecting cells from damage). For this review we choose to focus more on the abilities of hydroxycinnamic as a potential antioxidant, as this review is mostly focus on environmental stresses (biotic and abiotic stresses) and the potential role of hydroxycinnamic in conferring tolerance in this environmental stresses.
Abstract:
"Plant immobility renders them constantly susceptible to various abiotic and biotic stresses." - animal organisms are mobile and are also exposed to these factors.
We thank the reviewer for the comment. Yes we acknowledge the fact that although animals are mobile they are also impacted by these stresses, however the stresses are much more impactful to plants due to their immobility.
"Phenolic acid can be divided into different subclasses" - maybe acids?
We thank the reviewer for the comment. What we meant is that phenolic acids has sub groups which are hydroxycinnamic acids and hydrobenzoic acids.
Introduction:
Line 37: Are viruses biotic or abiotic?
We thank the reviewer for the comment. Viruses are considered abiotic stress since they’re non-living organism therefore we have rectify this in the review.
"Examples of secondary stresses includes osmotic stress and oxidative stress." - (include?) please add a reference to the literature
We thank the reviewer for the comment. We have since added the reference.
"antioxidant system such as ascorbate or glutathione" - maybe antioxidant compounds? Ascorbate is not a system.
We thank the reviewer for the comment. Ascobate is a compound that also plays a pivital role in a Asada pathway (also known as an ascorbate system) due to the fact that ascorbate peroxidase utilises ascorbate. This is a similar response with gluthathione
"In this review, we will highlight current studies on the diversity of hydroxycinnamic acid metabolites" - To make it more "current", I recommend enriching the bibliography with newer items. 30/71 are from the last 5 years and 43/71 are from the last 10 years.
We thank the reviewer for the comment. We have since updated the references.
"Phenolic acids are readily absorbed through the intestinal tract walls of
mammals, on a regular basis from vegetables, fruits, cereals, tea/coffee, and spices, which offer roughly 25 mg per day depending on the diet chart." - please provide a reference to the literature
We thank the reviewer for the comment. We have since added the reference.
"Phenolic acids generation in plants" - this chapter lacks the latest reports, facts that have been known for years are described. This chapter may be shortened to the necessary minimum in favor of more current research.
We thank the reviewer for the comment. We have took the reviewers advice and updated the reference accordingly
Salt stress - no link to current research: "Nedved, E. L., Kalatskaja, J. N., Ovchinnikov, I. A., Rybinskaya, E. I., Kraskouski, A. N., Nikalaichuk, V. V., ... & Laman, N. A. (2022). Growth parameters and antioxidant activity in cucumber seedlings with the application of chitosan and hydroxycinnamic acids conjugates under salt stress. Applied Biochemistry and Microbiology, 58(1), 69-76."
Drought stress - "Sarker, U., & Oba, S. (2018). Drought stress enhances nutritional and bioactive compounds, phenolic acids and antioxidant capacity of Amaranthus leafy vegetable. BMC Plant biology, 18(1), 1-15."
Biotic stress "Macoy, D. M. J., Uddin, S., Ahn, G., Peseth, S., Ryu, G. R., Cha, J. Y., ... & Kim, M. G. (2022). Effect of hydroxycinnamic acid amides, coumaroyl tyramine and coumaroyl tryptamine on biotic stress response in Arabidopsis. Journal of Plant Biology, 1-11."
Please analyze the rest of the text in terms of current available literature.
We thank the reviewer for the comment. We have took their advice and have since updated our references accordingly.
Materials and methods - no section, it is not known how the research for the literature review was planned. What queries were performed in the past and what filters were used?
We thank the reviewer for the comment. We followed a traditional way of writing a review and didn't do a systematic review hence the reason the is no materials and methods section. This review has been long in the making it took the important finding that came from multiple reviews that have been done from Plant OMICS lab, plant biotech laboratory as most of our work is based on small signaling molecules. This is the combination of multiples review. But focused on the important findings between the University of Zululand, University of the Western Cape, Free State University and the University of Stellenbosch.
Reviewer 2 Report
Comments and Suggestions for Authors
This review addresses a topical issue, the information provided reveals the potential of hydroxycinnamic acid metabolites/derivatives as potential antioxidants.
The evidence presented highlights the use of different secondary metabolites associated with hydroxycinnamic acid and their contribution to plant growth and development.
The introduction is robust, well considered, reflecting the current state of knowledge. The information included in this review is well-organized, presented in a well-structured manner.
The study provides an overview the diversity of hydroxycinnamic acid metabolites/derivatives as well as the possible role of enzyme inhibitors involved in the formation of these metabolites.
Potential technologies that could be used as key approaches for understanding the mechanism of interactions between these derivatives and ROS homeostasis were also explored.
The influence of phenolic acids involved in the phenylpropanoid pathway and their role under stress conditions are well detailed.
The figures are very well designed and representative for the topic.
The scientific value of this manuscript is not very high and its genuine contribution is not clearly highlighted to reflect its added value. Please, pay more attention to this issue and point out how this study contributes to the progress of knowledge.
Section 10. Conclusions and Future Perspectives should be separated into two parts: Future Perspectives and Conclusions. In the conclusions, please highlight the innovative aspects of the study.
The references are relevant to this topic, but are not adequately cited in the manuscript. Please refer to the journal requirements! Also, the References section is not written according to the requirements of the journal.
I recommend a major revision and reconsideration of the issues mentioned.

Minor editing of English language required.
Author Response
Resubmission of manuscript “Insights into the effects of hydroxycinnamic acid and its secondary metabolites as antioxidants for oxidative stress and plant growth under environmental stresses”
Thank you for the opportunity to revise our manuscript entitled ‘Insights into the effects of hydroxycinnamic acid and its secondary metabolites as antioxidants for oxidative stress and plant growth under environmental stresses’ We appreciate the careful review and constructive suggestions made by the two reviewers. It is our belief that the manuscript is substantially improved after making the suggested edits.
Following this letter we have addressed the reviewer comments with our responses in bold italics, including how and where the text was modified. Changes made in the manuscript are marked using track changes. The revision has been developed in consultation with all co-authors, and each author has given approval to the final form of this revision.
Thank you for your consideration and we look forward to your response.
Kind regards,
Dr. Mbukeni Nkomo
REVIEWER #2 COMMENTS:
Comments and Suggestions for Authors
This review addresses a topical issue, the information provided reveals the potential of hydroxycinnamic acid metabolites/derivatives as potential antioxidants.
The evidence presented highlights the use of different secondary metabolites associated with hydroxycinnamic acid and their contribution to plant growth and development.
The introduction is robust, well considered, reflecting the current state of knowledge. The information included in this review is well-organized, presented in a well-structured manner.
The study provides an overview the diversity of hydroxycinnamic acid metabolites/derivatives as well as the possible role of enzyme inhibitors involved in the formation of these metabolites.
Potential technologies that could be used as key approaches for understanding the mechanism of interactions between these derivatives and ROS homeostasis were also explored.
The influence of phenolic acids involved in the phenylpropanoid pathway and their role under stress conditions are well detailed.
The figures are very well designed and representative for the topic.
We thank the reviewer for the comment with regards to our figures.
The scientific value of this manuscript is not very high and its genuine contribution is not clearly highlighted to reflect its added value. Please, pay more attention to this issue and point out how this study contributes to the progress of knowledge.
We thank the reviewer for the comment. We have rework the review and also updated the work and references and clearly highlighted the contribution of knowledge to the paper.
Section 10. Conclusions and Future Perspectives should be separated into two parts: Future Perspectives and Conclusions. In the conclusions, please highlight the innovative aspects of the study.
We thank the reviewer for the comment. Have since took the advice in the manuscript.
The references are relevant to this topic, but are not adequately cited in the manuscript. Please refer to the journal requirements! Also, the References section is not written according to the requirements of the journal.
We thank the reviewer for the comment. Wehave since made the necessary changes in our manuscript.
Round 2
Reviewer 1 Report
Comments and Suggestions for Authors
The Authors have answered all suggestions.
Thank you for providing the corrected version. After reading I noticed many changes and they improved the quality of the manuscript. In the current version, it can be accepted after English proofreading.
Comments on the Quality of English Language
The article must be checked by a professional English reader. The Authors have not corrected English.
Reviewer 2 Report
Comments and Suggestions for Authors
The authors have addressed all suggestions and comments, and the manuscript has been improved accordingly. Therefore, it can be accepted for publication in this form.